# Learning under Label Proportions for Text Classification

**Jatin Chauhan**
UCLA
chauhanjatin100@cs.ucla.edu

**Xiaoxuan Wang**
UCLA
xw27@cs.ucla.edu

**Wei Wang**
UCLA
weiwang@cs.ucla.edu

## Abstract

We present one of the preliminary NLP works under the challenging setup of Learning from Label Proportions (LLP), where the data is provided in an aggregate form called bags and only the proportion of samples in each class as the ground truth. This setup is inline with the desired characteristics of training models under Privacy settings and Weakly supervision. By characterizing some irregularities of the most widely used baseline technique DLLP, we propose a novel formulation that is also robust. This is accompanied with a learnability result that provides a generalization bound under LLP. Combining this formulation with a self-supervised objective, our method achieves better results as compared to the baselines in almost $87\%$ of the experimental configurations which include large scale models for both long and short range texts across multiple metrics ([Code Link](Code Link)).

## 1 Introduction

The supervised classification setting in machine learning usually requires access to data samples with ground truth labels and our goal is to then learn a classifier that infers the label for new data points. However, in many scenarios obtaining such a dataset with labels for each individual sample can be infeasible and thus calls attention to alternative training paradigms. In this work, we study the setup where we are provided with *bags* of data points and the *proportion* of samples belonging to each class (instead of the one-one correspondence between a sample and its label). The inference however, still needs to be performed at the instance level in order to extract meaningful predictions from the model. This setup, which following (Yu et al., 2013) we refer to as *Learning from Label Proportions* (LLP), is attractive especially for at least two broad reasons (Nandy et al., 2022).

The first of these is **Privacy** centric learning (O'Brien et al., 2022). With ever increasing de-mand of user privacy in almost all applications of digital mediums, the individual record (here the label) for each sample (say a user's document) can't be exposed and thus learning in a fully supervised manner deems infeasible. The most notable applications include medical data (Yadav et al., 2016) and e-commerce data (O'Brien et al., 2022), amongst various others. Both of these contain abundant language data with multiple use cases of learning a classification model to perform disease prediction from patient's medical data and analyzing user's behavior for example, respectively. This makes LLP a highly relevant learning paradigm to train models, both large and small, for data that needs *k-anonymity*. The second relevant property is **Weak Supervision**. Since it is not always feasible to obtain clean data at a large scale, the training of the NLP models, both large and small, relies heavily on weakly supervised or self-supervised learning scrapped from the open web. LLP can play a key role as obtaining data at an *aggregated* level (formal definition in Section 2) can be a relatively easier and more feasible process.

While there exist various prior works that have proposed new formulations to learn under this setting (Musicant et al., 2007; Kuck and de Freitas, 2012; Tsai and Lin, 2019), many of them are either not readily applicable to learning deep models or very difficult to align with language data. Furthermore, to our knowledge, there exists only one prior work of (Ardehaly and Culotta, 2016) that discusses LLP for language data but is out of scope of this work as they focus on domain adaptation. We provide an elaborate discussion of the LLP literature in Section 5.

In this work, we address some shortcomings of one of the first loss formulations proposed for learning deep neural networks under LLP setup by (Ardehaly and Culotta, 2017), termed as DLLP method. For a given bag, DLLP optimizes the KL divergence between the ground truth bag level pro-

portion and the aggregate of the predictions for the instances in the entire bag. In Section 3, we highlight certain properties of DLLP objective that can be highly undesirable for training deep networks. Motivated by this, we propose a novel objective function that is a parametrization of the *Total Variation Distance* (TVD), which itself is a lower bound to the KL via the *Pinsker's inequality*. Our formulation enjoys more functional flexibility because of the introduced parameter while retaining the *outlier robustness* property of the TVD. We also discuss some theoretical results for the proposed novel formulation. Lastly, we combine our formulation with an auxiliary self-supervised objective that greatly aids in representation learning during the fine-tuning stage of the large scale NLP models experimented with.

Experimentally, we first demonstrate that the proposed formulation is indeed better and align with the theoretical motivation provided. In the main results, we demonstrate that our formulation achieves better results compared to the baselines in almost $87\%$ of the 20 extensive configurations across 4 widely used models and 5 datasets. We observe up to $40\%$ improvement in weighted precision metric on the BERT model. In many cases, the improvements range between $2\%$ to $9\%$ which are highly substantial. Further analysis including the bag sizes and hyperparameters also provide interesting insights into our formulation.

To summarize, we have the following contributions: (i) A novel loss formulation that addresses the shortcomings of the previous work with supporting theoretical and empirical results. (ii) One of the preliminary works discussing the application of LLP to natural language tasks. (iii) Strong empirical results demonstrating that our method outperforms the baselines in most of the configurations.

## 2   Preliminaries

We consider the standard multi-class classification setup with $C$ classes ($\mathcal{C} = [C] = \{1, ..., C\}$). The input instances $\mathbf{x}$ are sampled from an unknown distribution over the feature space $\mathcal{X}$. Contrary to the availability of the data samples of the form $(\mathbf{x}, \mathbf{y})$, where $\mathbf{y} \in \mathcal{C}$, to train the model in full supervision, we are provided a **bag** of input instances, $B = \{\mathbf{x}_i | i \in [|B|]\}$ ($|B|$ is the cardinality), with associated *proportions* of the labels $\rho = (\rho^1, \rho^2, ..., \rho^C)$. The elements of $\rho$ are defined

as:

$$\rho^j = \frac{|\{\mathbf{x}_i | \mathbf{x}_i \in B, \mathbf{y}_i = j\}|}{|B|} \qquad (1)$$

It is important to note that the ground truth labels $\mathbf{y}_i$ are not accessible and the entire information is contained in $\rho$. This $\rho$ essentially provides the *weak* supervision for training the model.

Given the training bags along with the label proportions as the tuple $(B_i, \rho_i)$, the objective is still to learn an *instance* level predictor that maps an instance $\mathbf{x}$ to the class distribution $\Delta = \{z \in \mathbb{R}_+^C | \sum_{l=1}^C z^l = 1\}$. The predicted class is attained as: $\arg\max_{c \in \mathcal{C}} \Delta$. We denote the classifier by $f_\theta : \mathcal{X} \to \Delta$ with learnable parameters $\theta \in \Theta$, which typically is a neural network such as BERT with appropriate classification head.

## 3   Method

Since the supervision under the LLP setup is coarse grained, given by $\rho$, the works in the literature (Ardehaly and Culotta, 2017), (Dulac-Arnold et al., 2019) utilize training criteria based on the true proportions $\rho_i$ and predicted proportions $\tilde{\rho}_i$ for the input bag $B_i$. The predicted proportions $\tilde{\rho}_i$ are typically some function of predicted class distributions $f_\theta(\mathbf{x}_j)$ for $\mathbf{x}_j \in B_i$ as $\tilde{\rho}_i = g(f_\theta(\mathbf{x}_1), ..., f_\theta(\mathbf{x}_{|B_i|}))$. The popular choice for $g$ is the *mean* function and we retain the same in this work. A descriptive pipeline of the LLP setup is provided in Figure 1.

### 3.1   Motivation

The work of (Ardehaly and Culotta, 2017) proposed one of the first loss formulations to learn deep neural networks under the LLP setup. Their loss objective over the proportions $\rho_i$ and $\tilde{\rho}_i$ is given by:

$$\mathcal{L}_{DLLP} = \sum_{c=1}^C \rho_i^c log \frac{\rho_i^c}{\tilde{\rho}_i^c}$$

.

**Definition 1** (Lipschitz Continuity)**.** A function $h$ is called Lipschitz continuous if there exists a positive constant $K$ such that $||h(x) - h(y)|| \leq K||x - y||, \forall x, y \in dom(h)$ .

We note some irregularities of $\mathcal{L}_{DLLP}$ which can lead to sub-optimal parameter values post training:

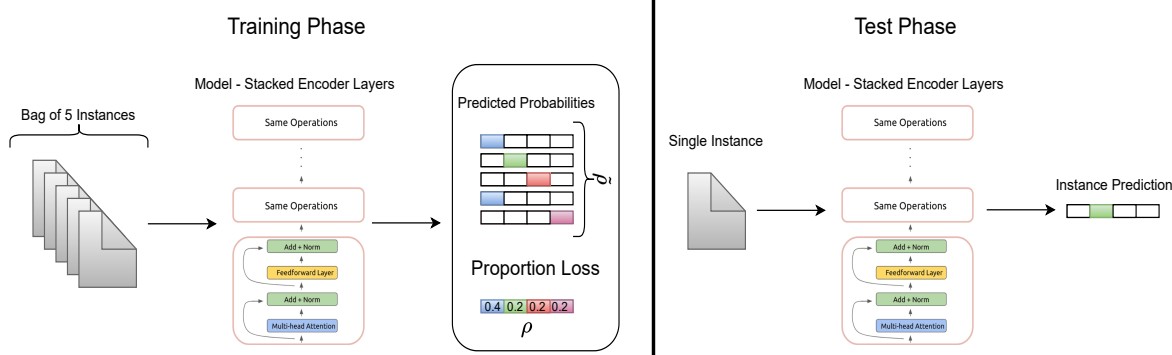

Figure 1: Demonstration of the Training (left) and Test (right) phase for LLP setup respectively. During training, a bag of instances is received and the output probabilities (a 4 class example setup, outputs colored based on the $\arg\max$ values) generated by the model are aggregated (computation of $\tilde{\rho}$). The proportion loss against the true proportion $\rho$ is then computed. In the test phase, the predictions are required to be performed at the instance level.

1. It is *not upper bounded*: Since the denominator of the log is the quantity $\tilde{\rho}_i^c$, which can be arbitrarily close to 0, the loss can go unbounded on the positive real axis.

2. It is not *robust*: follows from the previous point as small changes in $\rho^c$ and $\tilde{\rho}_i^c$ can lead to huge variations in the output.

3. It is not *symmetric*: due to the asymmetric nature of the definition of the objective.

4. It is not *Lipschitz continuous* in either of the arguments: follows from the first point.

5. It is not *Lipschitz smooth* (associating to gradients) in either of the arguments: stating that the gradients of the loss are not Lipschitz continuous. This can be associated with the largest eigenvalue of the Hessian, which exists for most losses in NN training and is critical to the training dynamics (Gilmer et al., 2022), thus dictating the optima, if achieved.

The aforementioned irregularities are undesirable especially for deep neural network optimization. This motivates us to propose a new loss function that partly remedies the potential failure modes of $\mathcal{L}_{DLLP}$. To begin, we state the following result:

**Lemma 1.** *[Pinsker's Inequality] For the probability mass functions $\rho$ and $\tilde{\rho}$,*

$$\mathcal{L}_{DLLP} = D_{KL}(\rho||\tilde{\rho}) \geq \frac{1}{2}\left(\sum_{c\in\mathcal{C}}|\rho^c - \tilde{\rho}^c|\right)^2$$
$$= 2 \times D_{TV}(\rho||\tilde{\rho})^2 \quad (2)$$

*where, $D_{KL}$ is the KL Divergence and $D_{TV}$ is the Total Variation Distance, $\rho^c$ as defined previously. $\rho^c$ is used as a shorthand notation over the domain here.*

### 3.2 Proposed Formulation

$D_{TV}$ has been utilized as a loss function in multiple works in the deep learning literature (Zhang et al., 2021; Knoblauch and Vomfell, 2020). While it admits a lower bound to the $\mathcal{L}_{DLLP}$ term, more importantly it is *robust* (Knoblauch and Vomfell, 2020). To further obtain higher functional flexibility while retaining this robustness property, we propose the following loss formulation, termed as $\mathcal{L}_{TV\star}^{\alpha}$ with hyperparameter $\alpha$:

$$\mathcal{L}_{TV\star}^{\alpha} = \frac{1}{2}\left(\sum_{c\in\mathcal{C}}|\rho^c - \tilde{\rho}^c|^{\alpha}\right)^{2/\alpha} \quad (3)$$

We discuss and prove various properties of $\mathcal{L}_{TV\star}^{\alpha}$ in Appendix A.2 and note that it does not admit any of the aforementioned irregularities stated in Section 3.1 for a broad range of values for $\alpha$ thus providing a better alternative. Further, we also empirically demonstrate the superiority of optimizing the model with the $\mathcal{L}_{TV\star}^{\alpha}$ against $\mathcal{L}_{DLLP}$ in Section 4.

### 3.3 Learnability under $\mathcal{L}_{TV\star}^{\alpha}$

We provide the following learnability result for a class of binary classifiers under the proposed formulation $\mathcal{L}_{TV\star}^{\alpha}$. The extension to multi-class setting is feasible via the use of Natarajan dimension and corresponding tools, however it is slightly more sophisticated and thus we leave it for future work.

**Definition 2** (VC Dimension). The maximum cardinality of a set $S = (\mathbf{x}_1, ... \mathbf{x}_m) \in \mathcal{X}^m$ of points that can be shattered by the hypothesis class $H$ under consideration.

**Theorem 2.** *Consider a hypothesis class $H$ containing functions $f : \mathcal{X} \rightarrow \{0,1\}$ and a target function $f_0 \in H$. Assume a sample $S$ with $m$ instances sampled i.i.d from $\mathcal{X}$. We represent the VC dimension of the class $H$ by $\mathcal{V}$. Further, denoting $\rho_f = \mathbb{E}_{\mathbf{x} \sim \mu} 1_{[f(\mathbf{x})=1]}$ where $\mu$ represents the probability measure over $\mathcal{X}$ and also $\tilde{\rho}_f = \frac{1}{m} \sum_{\mathbf{x} \in S} 1_{[f(\mathbf{x})=1]}$. Then with probability at least $1 - \delta$, $\forall f \in H$ and $\forall \alpha > 0$, we have*

$$\mathcal{L}_{TV\star}^{\alpha}(\rho_f, \rho_{f_0}) \leq \mathcal{L}_{TV\star}^{\alpha}(\tilde{\rho}_f, \tilde{\rho}_{f_0}) +$$
$$\kappa \left( \sqrt{\frac{8\mathcal{V}log(em/\mathcal{V})}{m}} + \sqrt{\frac{2log(4/\delta)}{m}} \right) \quad (4)$$

*where $\kappa = 2^{2/\alpha - 1}$.*

The proof of the theorem is delegated to the Appendix. This result characterizes that for any function $f$ in the hypothesis class, the loss $\mathcal{L}_{TV\star}^{\alpha}$ between the *population output aggregates* obtained from $f$ and the target function can be bounded via the corresponding *sample output aggregates*. We use the term aggregates to denote the mean over the indicator values $1_{[f(\mathbf{x})=1]}$. (Fish and Reyzin, 2017) provided one of the early results under the framework of *Learnable from Label Proportions*, where they provided an in-depth analysis for the specific case which corresponds to $\alpha = 1$ in our formulation.

### 3.4 Auxiliary Loss

While the theorem above provides guarantees for the performance over aggregated output, in order to further improve the quantitative performance of the model, we follow the success of various works in the deep learning literature that utilize auxiliary losses along with the primary training objective for improved generalization of the model (Trinh et al., 2018; Zhu et al., 2022; Rei and Yannakoudakis, 2017). This is directly tied to better representation learning for the model.

The category of *Self Supervised Contrastive* (SSC) losses has emerged as one of the most promising representation learning paradigms in recent years (Fang and Xie, 2022; Meng et al., 2022) and thus a popular choice for auxiliary losses as well. The recent work of (Nandy et al., 2022) proposed an SSC loss based on the embeddings of the instances

from the penultimate layer of the model. The highlight of their proposed approach is that it *does not require* explicit data augmentation or random sampling to construct negative pairs. This directly improves the runtime of the algorithm by a factor of 2, assuming linearity, as compared to many other popular choices of SSC losses (we refer the reader to the survey of (Jaiswal et al., 2020)). We therefore utilize their SSC objective to improve the overall performance of our formulation. Rewriting the model as $f_\theta = f^1(f^2(\mathbf{x}))$, where $f^1$ and $f^2$ are the parametrized sub-networks performing classification (only the final layer, ie, the classification head) and the representation learning respectively, the loss is given as

$$\mathcal{L}_{SSC} = \frac{1}{|B_i|} \sum_{\mathbf{x}_j \in B_i} -log \frac{e^{s(f^2(\mathbf{x}_j), f^2(\mathbf{x}_j))}}{\sum_{\mathbf{x}_k \in B_i} e^{s(f^2(\mathbf{x}_j), f^2(\mathbf{x}_k))}}$$
$$(5)$$

here $s$ is a similarity function between the embeddings, given by *cosine*, $s(a, b) = \frac{a^T b}{||a||_2 ||b||_2}$.

Thus the overall training objective for our method is the following objective:

$$\mathcal{L} = \mathcal{L}_{TV\star}^{\alpha} + \lambda \mathcal{L}_{SSC} \quad (6)$$

where $\lambda$ is the hyperparameter to control the auxiliary loss. The other hyperparameter in this objective is $\alpha$ under $\mathcal{L}_{TV\star}^{\alpha}$

## 4 Experiments

To investigate the effectiveness of the novel loss formulation in Eq. 6, we conduct extensive experiments across various models and datasets. These cover both short and long range texts as well as corresponding models developed to handle these texts. We begin by providing a brief overview of the setup.

### 4.1 Setup

**Models**: We consider the following 4 models: (i) *BERT* proposed by (Devlin et al., 2019) where we use a fully connected layer on the [CLS] token for classification. The documents are truncated at length of 256. (ii) *RoBERTa* proposed by (Liu et al., 2019b) where we use the similar setting as BERT to perform classification. (iii) *Longformer* proposed by (Beltagy et al., 2020) is explicitly designed to handle long range documents via an attention mechanism linear in the sequence length. The

input documents are truncated at the length of 2048. (iv) *DistilBert* proposed by (Sanh et al., 2019) is a distilled version of the primary BERT model with only around $40\%$ of the parameters but retaining up to $95\%$ performance. We truncate the sequences at the length of 512. All models are loaded using the Huggingface Transformers library (Wolf et al., 2019).

**Datasets**: We experiment on the following datasets: (i) *Hyperpartisan News Detection* (Kiesel et al., 2019), is a news dataset where the task is to classify an article having an extreme left or right wing standpoint. (ii) *IMDB* (Maas et al., 2011), is a movie review dataset for binary sentiment classification. (iii) *Medical Questions Pairs* (Medical QP) (McCreery et al., 2020), contains pairs of questions where the task is to identify if the pair is similar or distinct. Following the common practice, we concatenate the questions in a given pair before tokenizing. (iv) *Rotten Tomatoes* (Pang and Lee, 2005), is a movie reviews dataset where each review is classified as positive or negative. Along with these, we also include a multi-class (non-binary) dataset *Financial Phrasebank* (Malo et al., 2014) with 3 classes to assess the formulations under a more difficult setting. It is a sentiment dataset of sentences written by various financial journalists from financial news. The statistics of the datasets are provided in Table 5.

**Training**: We fine tune the pretrained versions of the 4 models under the LLP setup over the bags. The input to the models follows the structure of (input, label), where "input" is a bag $B_i$ and "label" is the ground truth proportion $\rho_i$. To construct the bags $B_i$ during training, we gather the instances in groups based on the size mentioned in Tables 6 and 7, from the list of all training instances. To obtain the corresponding ground truth label proportions per bag $\rho_i$, we directly average the one-hot vector representations of the labels for the instances in the bag. Note that the ground truth labels *are not used* at any other phase of the setup except testing which is performed at the instance level. There, the use of the ground truth labels is restricted to compare the performance of our formulation against the baselines. We construct new bags at each epoch by shuffling the data. This is inline with practical purposes where an instance can be shared across different bags across different training epochs, for example - advertising (CTR prediction) and medical records. The setup with fixed bags beforehand

is much harder and potentially requires extremely large sized datasets (Nandy et al., 2022). Adhering to the memory constraints, we utilize one bag per iteration typically containing 16 - 32 instances (Tables 6 and 7) which is the standard batch size range for fine-tuning. Although, there is no strict constraint for the number of bags per iteration and more bags can be utilized given sufficient memory. Most of the fine-tuning details and parts of code are retained from the recent work of (Park et al., 2022) which provides an extensive comparison of large scale models for standard supervised classification. Since we don't assume access to true labels even for the validation set, common techniques for parameter selection (based on performance on the validation set) are difficult to utilize. Following (Nandy et al., 2022), we thus use an aggregation of both training and validation losses across the last few epochs to tune the hyperparameters. Tables 6 and 7 detail the important hyperparameters. All the experiments were conducted on a single NVIDIA - A100 GPU.

**Baselines**: While there are no prior works directly studying the setup of LLP for natural language classification tasks, we consider two existing approaches that can be readily applied here. The first is the formulation of (Ardehaly and Culotta, 2017) which we refer as *DLLP* and the corresponding loss as $\mathcal{L}_{DLLP}$, presented in Section 3. The second is the work by (Nandy et al., 2022) which uses the $\mathcal{L}_{DLLP}$ loss in conjunction with the contrastive auxiliary objective in Eq. 5, which we will refer as *LLPSelfCLR* baseline. They also propose an additional diversity penalty to further diversify the learned representations. Section 5 provides an elaborate overview of other works in the literature.

We use the **weighted** versions of *precision, recall and F1* score to evaluate the performance of the models as these metrics quantify the performance in a holistic and balanced manner. The acronyms *W-P*, *W-R* and *W-F1* for these metrics are used while describing the results. We perform the analysis experiments on diverse configurations in order to ensure coverage and completeness across the possible combinations.

## 4.2 Main Results

Table 1 highlights the main results of our complete loss formulation in Eq. 6 and comparison against the baselines on the binary classification datasets. We also quantify the difficulty of the LLP setup

| Base Model | Formulation | Dataset | | | | | | | | | | | |
|---|---|---|---|---|---|---|---|---|---|---|---|---|---|
| | | Hyperpartisan | | | IMDB | | | Rotten Tomatoes | | | Medical QP | | |
| | | W-P | W-R | W-F1 | W-P | W-R | W-F1 | W-P | W-R | W-F1 | W-P | W-R | W-F1 |
| Longformer | DLLP | 34.17 | 58.46 | 43.13 | 86.06 | 85.89 | 85.87 | 64.66 | 55.25 | 46.71 | 51.45 | 50.41 | 40.41 |
| | LLPSelfCLR | **79.94** | 75.38 | 72.99 | 90.39 | 90.38 | 90.38 | 83.87 | 83.42 | 83.37 | 53.18 | **53.03** | 52.55 |
| | Ours | 79.92 | **80.00** | **79.94** | **92.49** | **94.48** | **92.47** | **85.61** | **85.07** | **85.01** | **53.19** | **53.03** | 52.52 |
| | Oracle | 95.72 | 95.38 | 95.33 | 95.49 | 95.48 | 95.48 | 88.63 | 88.59 | 88.58 | 86.91 | 86.37 | 86.32 |
| RoBERTa | DLLP | 34.17 | 58.46 | 43.13 | 89.03 | 89.01 | 89.01 | 78.85 | 69.29 | 66.39 | 25.08 | 50.08 | 33.42 |
| | LLPSelfCLR | 54.34 | 53.84 | 54.04 | 90.08 | 90.08 | 90.08 | 77.42 | 77.23 | 77.19 | 50.47 | 49.18 | 49.31 |
| | Ours | **61.06** | **61.53** | **56.21** | **90.14** | **90.13** | **90.13** | **85.41** | **85.30** | **85.29** | **53.70** | **53.53** | **53.03** |
| | Oracle | 69.61 | 64.61 | 64.39 | 93.58 | 93.58 | 93.57 | 88.47 | 88.45 | 88.44 | 83.62 | 82.92 | 82.83 |
| BERT | DLLP | 55.10 | 58.46 | 45.69 | 84.33 | 84.33 | 84.33 | 75.04 | 74.13 | 73.89 | 56.36 | 56.32 | 56.22 |
| | LLPSelfCLR | 50.71 | 56.92 | 46.95 | 86.85 | 86.82 | 86.83 | 75.04 | 74.13 | 73.89 | **66.08** | **66.00** | **65.96** |
| | Ours | **77.36** | **63.07** | **52.73** | **87.58** | **87.50** | **87.49** | 74.94 | 73.80 | 73.49 | 64.54 | 64.53 | 64.52 |
| | Oracle | 87.70 | 87.69 | 87.61 | 91.59 | 91.57 | 91.57 | 86.00 | 85.86 | 85.85 | 81.44 | 81.28 | 81.25 |
| DistilBERT | DLLP | 34.17 | 58.46 | 43.13 | 86.89 | 86.86 | 86.85 | 79.63 | 77.41 | 76.98 | 25.08 | 50.08 | 33.42 |
| | LLPSelfCLR | 67.92 | 67.69 | 65.60 | 86.89 | 86.86 | 86.85 | 79.63 | 77.41 | 76.98 | 52.73 | 52.70 | 52.63 |
| | Ours | **70.21** | **69.23** | **66.93** | **87.64** | **87.62** | **87.62** | **79.69** | **77.46** | **77.03** | **55.17** | **55.17** | **55.17** |
| | Oracle | 92.56 | 92.30 | 92.22 | 92.81 | 92.82 | 92.81 | 83.76 | 83.75 | 83.75 | 77.83 | 76.35 | 76.04 |

Table 1: Comparison of our formulation against the baseline methods DLLP and LLPSelfCLR. *Oracle* denotes the performance in the supervised setting, the bag size of 1, and is demonstrated solely to quantify the difficulty of the problem. The best numbers are highlighted in bold and second best underlined. Our formulation achieves better results in almost 83% of the configurations.

| Formulation | Longformer | | | RoBERTa | | |
|---|---|---|---|---|---|---|
| | W-P | W-R | W-F1 | W-P | W-R | W-F1 |
| DLLP | 69.97 | 70.18 | 65.85 | 76.97 | 73.91 | 72.44 |
| LLPSelfCLR | 80.62 | 78.57 | 78.30 | 78.62 | 75.67 | 74.18 |
| Ours | **81.38** | **79.50** | **79.06** | **80.64** | **77.43** | **76.82** |
| Oracle | 82.69 | 80.53 | 80.43 | 82.64 | 80.84 | 81.18 |
| | BERT | | | DistilBERT | | |
| | W-P | W-R | W-F1 | W-P | W-R | W-F1 |
| DLLP | 63.32 | 61.90 | 58.91 | 74.14 | 68.42 | 63.92 |
| LLPSelfCLR | 73.92 | 70.80 | 70.96 | 72.43 | 73.08 | 71.80 |
| Ours | **78.02** | **75.67** | **73.88** | **76.58** | **75.46** | **72.75** |
| Oracle | 79.67 | 77.01 | 76.51 | 80.90 | 77.63 | 78.12 |

Table 2: Comparison of our formulation against the baselines on Financial Phrasebank dataset. The notations follow same meaning as described previously. Our formulation achieves better results across all the models.

by providing the results in the corresponding fully supervised setup, with bag size of 1, named as *Oracle*. Note that Oracle is not a true feasible baseline here. The same result for DLLP and LLPSelfCLR in some configurations in the table correspond to the cases where LLPSelfCLR does not outperform DLLP baseline and the optimal choice for its hyperparameter controlling the contrastive loss is almost 0. Same results for the DLLP baseline across the models such as RoBERTa, Longformer and Distil-BERT on Hyperpartisan dataset, demonstrate that it is not able to optimize and learn properly. Similar observations are accounted for Medical QP dataset. We see that our formulation achieves the best results in almost 83% of the values in the table cumulating across all configurations in the precision, recall and F1 scores. In the settings where we don't outperform the baselines, the relative margins are extremely small, often in the order of the decimal

precision. On the other hand, in the settings where we outperform the baselines, the relative gains are as large as up to 40% (BERT on Hyperpartisan dataset) and up to 10% (RoBERTa on Rotten Tomatoes and Hyperpartisan datasets) in weighted precision. Weighted Recall and F1-score mostly show similar gains in the corresponding configurations. In various configurations including DistilBERT and RoBERTa on Medical QP, Longformer on IMDB and Rotten Tomatoes), we also observe notable improvements ranging from 2% to 9% uniformly across the metrics. These results also align with the architectural designs and findings of previous works as Longformer achieves comparatively better performance especially on long-range datasets: Hyperpartisan and IMDB while DistilBERT exhibits more robust results compared to BERT. It is also important to note that the margins observed for our formulation are significant for the relatively small sized dataset Hyperpartisan which also generalizes to Medical QP to an extent. To summarize, the gains exhibited by our formulation are consistent across all models and datasets including small and large sized datasets with both long and short texts. The results for the Financial Phrasebank dataset are provided in Table 2. Similar to the results in the binary dataset configurations, we again observe that our formulation achieves consistently better results across the 4 models. Although the gains are not as high as 40% which were observed in some binary settings, we still note significant improvements of around 23% w.r.t to DLLP and 5.5% w.r.t

| Dataset | | W-P | W-R | W-F1 |
|---|---|---|---|---|
| Hyperpartisan | DLLP | 34.17 | 58.46 | 43.13 |
| | $\mathcal{L}_{TV\star}^{\alpha}$ | 76.80 | 61.53 | 49.72 |
| | Overall | 79.92 | 80.00 | 79.94 |
| Rotten Tomatoes | DLLP | 64.66 | 55.25 | 46.71 |
| | $\mathcal{L}_{TV\star}^{\alpha}$ | 84.37 | 84.32 | 84.31 |
| | Overall | 85.61 | 85.07 | 85.01 |

Table 3: Comparison of DLLP loss against our proposed $\mathcal{L}_{TV\star}^{\alpha}$ in Eq. 3 (note this is *without* the auxiliary loss of Eq. 5) for Longformer. Performance of the overall loss , Eq. 6, is also provided.

LLPSelfCLR on W-P for BERT model. Generally speaking, we observe around $2-3\%$ gains on the metrics across the models. These relative margins can be attributed to the fact that the multi-class setting with more than 2 classes is more challenging in both theory and practice.

### 4.3 Analysis

#### 4.3.1 Justification of $\mathcal{L}_{TV\star}^{\alpha}$ and $\mathcal{L}_{SSC}$

As stated in Section 3, the DLLP loss admits various undesirable properties which motivated us to propose the first part of the loss function in Eq. 6. Here, we provide an empirical evidence that $\mathcal{L}_{TV\star}^{\alpha}$(Eq. 2) indeed generalizes better than $\mathcal{L}_{DLLP}$ under various configurations. Table 3 provides the results comparing the two formulations on Longformer model over Hyperpartisan and Rotten Tomatoes datasets. It is evident that our formulation achieves significantly better results of which particularly noteworthy are the metric values of W-P on Hyperpartisan, more than **2x**, and W-F1 on Rotten tomatoes, almost **1.8x**. Other values also highlight similar trends and noteworthy margins. More experiments are provided in appendix A.1.1. Correspondingly, we can also observe the practical benefits of using SSC objective by comparing the results of $\mathcal{L}_{TV\star}^{\alpha}$ against the overall loss formulation of Eq. 6 in Table 3. We note that for Hyperpartisan, the relative improvements brought via the SSC objective are more notable in the W-F1 score. On the other hand, for Rotten Tomatoes, we observe only marginal improvements using SSC. This demonstrates that while $\mathcal{L}_{TV\star}^{\alpha}$ leads to strong results, using an auxiliary objective such as an SSC loss can further improve the performance, notably or marginally.

#### 4.3.2 Variation of Bag size

The size of the input bags is a direct control factor for the performance in the LLP setup. We thus conduct an experiment to note the performance of our method against the baselines for varying bag sizes. Previous works (Liu et al., 2019a; Dulac-Arnold et al., 2019) have exhaustively demonstrated that the performance of the models degrade as the bag size increases. This is due to *reduction* in the supervision available to train the models. Figure 2(a) plots the comparison for the different formulations for RoBERTa on Rotten Tomatoes dataset. These results align with the findings in the literature accounting for the amount of supervision provided. It is noteworthy that our method is more stable as compared to the baselines. For smaller bag sizes (2 - 8), all the models demonstrate fairly competitive performance. For the size 16 DLLP performs closer to our formulation however LLPSelfCLR incurs non-trivial degradation. For larger bag size, the performance reduces further (significantly for DLLP) and the gap between our formulation to the baselines admit large difference. This result is thus inline with the claims presented for our formulation in Section 3.

#### 4.3.3 Variation of hyperparameters

To analyze the effect of the hyperparameters on the performance, we conduct experiments by varying the values of $\alpha$ and $\lambda$, Eq. 6. While varying the value of one, the other is kept constant. The results are plotted in Figures 2(b) and 2(c). The analysis for $\alpha$ is performed over BERT on Medical QP and for $\lambda$ over BERT on IMDB to ensure diverse coverage. For both the subfigures corresponding to $\alpha$ and $\lambda$, it is non-trivial to find a direct trend. This aligns with many works in various machine learning literatures since the hyperparameters play a key role in determining the output performance and a small variation can account for large deviations in the model performance. In both the configurations, we first observe the optimal performance point as we increase the value of the corresponding hyperparameter followed by a sharp degradation and further improvement. However, it is noteworthy that for almost all the cases, all the 3 metrics provide highly calibrated results (ie, highly balanced results) across the y-axis, thus further verifying the efficacy of our approach.

### 5 Related Work

We discuss the literature of LLP via the following taxonomy:
**Shallow Learning based**: This is the category of models that doesn't leverage deep learning models. The notable works include (Musicant et al., 2007)

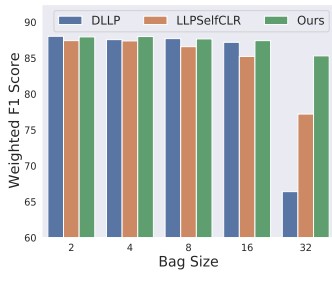
(a) Variation of Bag Size

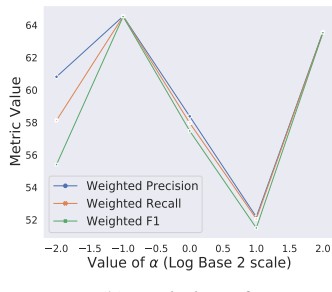
(b) Variation of $\alpha$

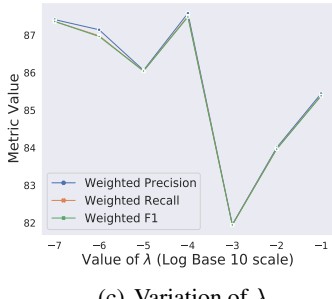
(c) Variation of $\lambda$

Figure 2: Variation of Bag size (subfigure a): For increasing bag size, we observe substantial difference between our method and the baseline formulations. Analysis of Hyperparameters $\alpha$ (subfigure b) and $\lambda$ (subfigure c): We don't observe a direct trend however the performance is well calibrated across the metrics.

(using SVM, k-NN), (Kuck and de Freitas, 2012; Hernández-González et al., 2013) (probabilistic models), (Scott and Zhang, 2020) (mutual contamination framework) etc. Note that these methods do not scale for higher dimensional and larger datasets, thus making these unfit for language tasks.

**Deep Learning based**: Some of the notable works include (Ardehaly and Culotta, 2017) and (Dulac-Arnold et al., 2019) which aim to minimize a divergence between the true proportions $\rho$ and the predicted proportions of a given bag $\tilde{\rho}$. (Dulac-Arnold et al., 2019) also used the concept of *pseudo labeling* (Lee, 2013), where an estimate of the individual labels within each bag is jointly optimized with the model during training. (Liu et al., 2021) revisited pseudo labeling for LLP by proposing a two-stage scheme that alternately updates the network parameters and the pseudo labels. However, both these works leveraging pseudo labeling perform experiments with computer vision datasets only. Furthermore, open source codes could not be found thus making it difficult to compare against these formulations.

**Auxiliary Loss based**: Recent trends in learning with auxiliary losses alongside a primary objective also gained attention in the LLP community. (Liu et al., 2019a) proposed LLP-GAN framework by incorporating generative adversarial networks (GAN). (Tsai and Lin, 2019) proposed LLP-VAT that incorporated an additional consistency regularizer to learn consistent representations in the setting where the input instances are perturbed adversarially. It is noteworthy that both these methods leverage vision datasets for experiments. The tasks of generation and adversarial perturbation incur significant overhead for NLP and in various cases,

this overhead can be larger than the primary objective task itself. For example, the generation of new instances via GANs. (Nandy et al., 2022) proposed a contrastive self-supervised auxiliary loss to improve the representation learning of the underlying deep network in conjunction with the DLLP loss. They augment this with an extra penalty term to further diversify the representations learned.

Lastly, to our knowledge, the only work that emphasizes the setting of LLP for natural language classification tasks is the work by (Ardehaly and Culotta, 2016). However, it is out of the scope of this work as they aim to improve the performance of models under the LLP setup when the training and testing data admit change in the underlying domain distributions, ie, the Domain adaptation setting.

# 6 Conclusion

We study the setup of learning under label proportions for natural language classification tasks. The LLP setup admits two properties: Privacy centric learning and Weak Supervision that render it as a highly practical modern research topic. We first discuss the shortcomings of one of the first LLP loss formulations proposed for learning deep neural networks. These provide the motivation for our novel loss formulation, termed as $\mathcal{L}^{\alpha}_{TV\star}$, that does not admit such irregularities. The formulation is also supported with theoretical justification and hardness results. Empirically, we justify that the new loss function achieves better performance on diverse configurations. The main results achieved by combining this loss with a self-supervised contrastive formulation achieve significantly better performance as compared to the baselines.

# 7 Limitations

While the proposed formulation achieves better results in the majority of the configurations, the gap between the best results and the corresponding supervised oracle is large in some cases. This can be configuration dependent and thus harder to interpret. Another limitation which follows from the black-box nature of the large models is that the current theoretical understanding of the proposed formulation are not sufficient especially at the interplay between $\mathcal{L}_{TV\star}$ and $\mathcal{L}_{SSC}$ in eq 6. More work in this direction will help unfold not only the underlying mechanisms but also aid in designing better loss formulations.

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

| Dataset | | W-P | W-R | W-F1 |
|---|---|---|---|---|
| IMDB | DLLP | 86.06 | 85.89 | 85.87 |
| | $\mathcal{L}_{TV\star}^{\alpha}$ | 88.40 | 88.36 | 88.36 |
| | Overall | 92.49 | 94.48 | 92.47 |
| Financial Phrasebank | DLLP | 69.97 | 70.18 | 65.85 |
| | $\mathcal{L}_{TV\star}^{\alpha}$ | 76.36 | 74.94 | 71.48 |
| | Overall | 81.38 | 79.50 | 79.06 |

Table 4: Comparison of DLLP loss against our proposed $\mathcal{L}_{TV\star}^{\alpha}$ in Eq 3 (note this is *without* the auxiliary loss of Eq 5) for Longformer. Performance of the overall loss , Eq 6, is also provided.

# A   Appendix

## A.1   Additional Experiments

This section contains additional experiments to support the proposed loss formulation as well as further analyse the hyperparameter variation. The diverse selection of model and datasets in the following experiments provide sufficient coverage across the configurations.

### A.1.1   Analysis of $\mathcal{L}_{TV\star}^{\alpha}$

Table 4 reports the comparison of DLLP against $\mathcal{L}_{TV\star}^{\alpha}$ on more datasets over longformer model. We note that the findings from table 3 extend to other configurations as well.

### A.1.2   Variation of hyperparameters

The variation of $\alpha$ value has been plotted in figure 3(a) for BERT on Hyperpartisan dataset.
The variation of $\lambda$ value has been plotted in figure 3(b) for RoBERTa on Financial Phrasebank dataset.

## A.2   Properties of $\mathcal{L}_{TV\star}^{\alpha}$

Here, we discuss the theoretical properties that the proposed loss formulation $\mathcal{L}_{TV\star}^{\alpha}$ admits. The following lemmas demonstrate that it does not exhibit any of the irregularities highlighted in Section 3.1 for a broad range of values of $\alpha$ used in majority of experiments .

**Lemma 3.** $\mathcal{L}_{TV\star}^{\alpha}$ *admits an absolute upper bound for all inputs pairs $\rho$ and $\tilde{\rho}$ for $\alpha \geq 1$.*

*Proof.* For $\alpha \geq 1$, using that $D_{TV} \leq 1$, we have that $\mathcal{L}_{TV\star}^{\alpha} \leq 2 \times D_{TV}^2 \leq 2$ □

**Lemma 4.** $\mathcal{L}_{TV\star}^{\alpha}$ *is symmetric.*

*Proof.* By definition of $\mathcal{L}_{TV\star}^{\alpha}$, we have $\mathcal{L}_{TV\star}^{\alpha}(\rho, \tilde{\rho}) = \mathcal{L}_{TV\star}^{\alpha}(\tilde{\rho}, \rho)$ □

**Lemma 5.** $\mathcal{L}_{TV\star}^{\alpha}$ *is Lipschitz continuous in both the arguments for $\alpha \geq 1$.*

*Proof.* Given the two arguments as discrete distributions $\rho_1, \rho_2$, we prove the result for $\rho_1$ and by symmetry it follows in both arguments.
We have that $\mathcal{L}_{TV\star}^{\alpha}(\rho_1, \rho_2)$ is differentiable almost everywhere in the interior of the simplex. We compute the gradient of $\mathcal{L}_{TV\star}^{\alpha}(\rho_1, \rho_2)$ w.r.t to $\rho_1^i$, corresponding to the $i^{th}$ element in the support,

$$\frac{\partial \mathcal{L}_{\mathcal{TV}\star}^{\alpha}(\rho_1, \rho_2)}{\partial \rho_1^i} = \left( \sum_i |\rho_1^i - \rho_2^i|^{\alpha} \right)^{\frac{2}{\alpha}-1}$$
$$\times |\rho_1^i - \rho_2^i|^{\alpha-1} \times sign(\rho_1^i - \rho_2^i) \tag{7}$$

Here, $sign(\cdot)$ is the sign function.
To show Lipschitz continuity, it is sufficient to show that the norm of the gradient, $||\nabla \mathcal{L}_{TV\star}^{\alpha}(\rho_1, \rho_2)||$, is bounded.
Now,

$$||\nabla \mathcal{L}_{TV\star}^{\alpha}(\rho_1, \rho_2)|| = \frac{\left( \sum_i |\rho_1^i - \rho_2^i|^{\alpha} \right)^{\frac{2}{\alpha}-1}}{\sqrt{\sum_i |\rho_1^i - \rho_2^i|^{2\alpha-2}}} \tag{8}$$

Given a fixed number of classes, it is trivial that

$$||\nabla \mathcal{L}_{TV\star}^{\alpha}(\rho_1, \rho_2)|| = \mathcal{O}(\kappa(C)) \tag{9}$$

Here, $\kappa(C)$ is a constant function of the number of classes. Thus, Lipschitz continuity holds.

□

**Lemma 6.** $\mathcal{L}_{TV\star}^{\alpha}$ *is Lipschitz smooth in both the arguments for $\alpha \geq 1$.*

*Proof.* Following the previous lemma, given the two arguments as discrete distributions $\rho_1, \rho_2$, we prove the result for $\rho_1$ and by symmetry it follows in both arguments.
We begin by noting that $\mathcal{L}_{TV\star}^{\alpha}$ is a convex function in either of the arguments due to the convexity of norms.
Furthermore, we also have

$$c_1 D_{TV}^2 \leq \mathcal{L}_{TV\star}^{\alpha} \leq c_2 D_{TV}^2 \tag{10}$$

for constants $c_1$ and $c_2$. This is obtained using equivalence of norms.

To show the desired smoothness result, it is sufficient to prove that the function

$$f(\rho_1) = \frac{B}{2} \rho_1^T \rho_1 - \mathcal{L}_{TV\star}^{\alpha}(\rho_1, \rho_2) \tag{11}$$

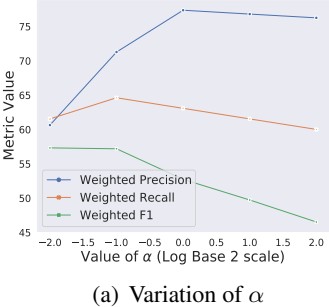

(a) Variation of $\alpha$

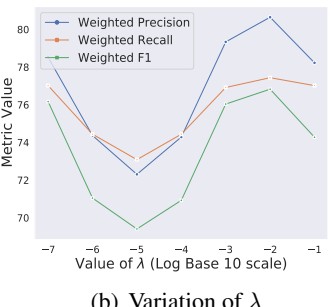

(b) Variation of $\lambda$

Figure 3: Variation of hyperparameters.

is convex $\forall \rho_1$ and fixed $\rho_2$. $B$ being the smoothness constant.

The key element to the proof is the convexity of the following

$$\frac{B_{c_1}}{2}\rho_1^T \rho_1 - c_1 D_{TV}^2(\rho_1, \rho_2) \qquad (12)$$

$B_{c_1}$ being the smoothness constant, which depends upon the fixed constant $c_1$ stated in Eq 10.

Thus, we can upper bound Eq 11 using Eq 12 to obtain the desired Lipschitz smoothness $\forall \alpha \geq 1$ .

$\square$

### A.3 Proof of Theorem 2

**Definition 3** (Empirical Rademacher Complexity). For a sample $S = (\mathbf{x}_1, ... \mathbf{x}_m) \in \mathcal{X}^m$ of points and a function class $\mathcal{A}$ of real valued functions, $A : \mathcal{X} \to \mathbb{R}$, the empirical Rademacher Complexity of $\mathcal{A}$ given $S$ is

$$\mathcal{R}_S = \frac{1}{m}\mathbb{E}_\sigma \left[ \sup_{A \in \mathcal{A}} \sum_{i=1}^m \sigma_i A(\mathbf{x}_i) \right] \qquad (13)$$

where $\sigma_i$ are independent random variables drawn from the Rademacher distribution, ie, $Pr(\sigma_i = +1) = Pr(\sigma_i = -1) = \frac{1}{2}$

*Proof of Theorem 2.* We begin using the equivalent bounds in PAC learning by (Shalev-Shwartz and Ben-David, 2014) and using the proof sketch of (Fish and Reyzin, 2017).
First, consider the loss function $l(f, x) = (f(x) - f_0(x))^2$. This is a function with $|l(f, x)| \leq 1$, thus a generalization bound can be derived using the *empirical Rademacher Complexity* $\mathcal{R}_S$ of the class $H$, stating that with probability at least $1 - \delta$, $\forall f \in$

$H$, we have:

$$\mathbb{E}_{\mathbf{x}\sim\mu}[l(f,x)] - \frac{1}{m}\sum_{x \in S} l(f,x) \leq \sqrt{\frac{2log(2/\delta)}{m}} +$$
$$2\mathbb{E}_{S\sim\mu^m}[\mathcal{R}_S(\{(l(f,x_1), ..., l(f,x_m))|f \in H\})] \qquad (14)$$

Using the Sauer-Shelah lemma we have that

$$|\{(f(x_1), ..., f(x_m))|f \in H\}| \leq \left(\frac{em}{\mathcal{V}}\right)^{\mathcal{V}} \qquad (15)$$

and we also have

$$|\{(l(f,x_1), ..., l(f,x_m))|f \in H\}| \leq \left(\frac{em}{\mathcal{V}}\right)^{\mathcal{V}} \qquad (16)$$

Since $|l(f, x) \leq 1|$ the Massart's lemma implies that for a sample $S$, we have the following

$$\mathcal{R}_S(\{(l(f,x_1), ..., l(f,x_m))|f \in H\}) \leq \sqrt{\frac{2\mathcal{V}log(em/\mathcal{V})}{m}} \qquad (17)$$

Thus combining Eq 14, 15, 16 and 17 above, for the loss $l$ we have with probability $1 - \delta/2$, $\forall f \in H$

$$\mathbb{E}_{\mathbf{x}\sim\mu}[l(f,x)] - \frac{1}{m}\sum_{x \in S} l(f,x) \leq$$
$$2\sqrt{\frac{2\mathcal{V}log(em/\mathcal{V})}{m}} + \sqrt{\frac{2log(4/\delta)}{m}} \qquad (18)$$

Repeating the argument, we also have with probability $1 - \delta/2$, $\forall f \in H$

$$\frac{1}{m}\sum_{x \in S} l(f,x) - \mathbb{E}_{\mathbf{x}\sim\mu}[l(f,x)] \leq$$
$$2\sqrt{\frac{2\mathcal{V}log(em/\mathcal{V})}{m}} + \sqrt{\frac{2log(4/\delta)}{m}} \qquad (19)$$

Thus using the union bound we obtain with probability $1 - \delta$, $\forall f \in H$

$$|\mathbb{E}_{\mathbf{x} \sim \mu}[l(f,x)] - \frac{1}{m}\sum_{x \in S} l(f,x)| \leq$$

$$2\sqrt{\frac{2\mathcal{V}log(em/\mathcal{V})}{m}} + \sqrt{\frac{2log(4/\delta)}{m}} \qquad (20)$$

Using Jensen's inequality and for any $\alpha > 0$ we also have the following ,

$$\mathbb{E}_{\mathbf{x} \sim \mu}[l(f,x)] \geq (\rho_f - \rho_{f_0})^2$$

$$= \left[\frac{1}{2} \times (|\rho_f - \rho_{f_0}|^\alpha + |(1 - \rho_f) - (1 - \rho_{f_0})|^\alpha)\right]^{2/\alpha} \qquad (21)$$

$$= 2^{1-2/\alpha}\mathcal{L}^\alpha_{TV\star}(\rho_f, \rho_{f_0}) \qquad (22)$$

Similarly, we also obtain

$$\frac{1}{m}\sum_{x \in S} l(f,x) \geq (\tilde{\rho}_f - \tilde{\rho}_{f_0})^2$$

$$= \left[\frac{1}{2} \times (|\tilde{\rho}_f - \tilde{\rho}_{f_0}|^\alpha + |(1 - \tilde{\rho}_f) - (1 - \tilde{\rho}_{f_0})|^\alpha)\right]^{2/\alpha} \qquad (23)$$

$$= 2^{1-2/\alpha}\mathcal{L}^\alpha_{TV\star}(\tilde{\rho}_f, \tilde{\rho}_{f_0}) \qquad (24)$$

Under a mild monotonicity condition, we can combine Eq 20, 22 and 24 to obtain the desired bound of Eq 4

$$\mathcal{L}^\alpha_{TV\star}(\rho_f, \rho_{f_0}) \leq \mathcal{L}^\alpha_{TV\star}(\tilde{\rho}_f, \tilde{\rho}_{f_0}) +$$

$$\kappa\left(\sqrt{\frac{8\mathcal{V}log(em/\mathcal{V})}{m}} + \sqrt{\frac{2log(4/\delta)}{m}}\right) \qquad (25)$$

where $\kappa = 2^{2/\alpha-1}$

$\square$

| Dataset | # Train | # Val | # Test | Average # Words |
|---|---|---|---|---|
| Hyperpartisan | 516 | 64 | 65 | 565 |
| IMDB | 22500 | 2500 | 25000 | 231 |
| Rotten Tomatoes | 7464 | 1068 | 2130 | 21 |
| Medical QP | 2134 | 305 | 609 | 40 |
| Fin. Phrasebank | 3394 | 486 | 966 | 23 |

Table 5: Statistics of the datasets. The first 3 columns represent total number of instances. Bag sizes in training and validation are provided in Table 6

| Model | Dataset | | | | | | | | | | | | | | |
|---|---|---|---|---|---|---|---|---|---|---|---|---|---|---|---|
| | Hyperpartisan | | | | | IMDB | | | | | Rotten Tomatoes | | | | |
| | B | E | LR | $\lambda$ | $\alpha$ | B | E | LR | $\lambda$ | $\alpha$ | B | E | LR | $\lambda$ | $\alpha$ |
| Longformer | 16 | 20 | $5e^{-5}$ | 1.0 | 1.0 | 32 | 10 | $3e^{-3}$ | 1.0 | 0.5 | 32 | 10 | $2e^{-4}$ | 1.0 | 0.33 |
| RoBERTa | 16 | 10 | $3e^{-3}$ | 1.0 | 2.5 | 32 | 10 | $3e^{-3}$ | 0.0 | 1.0 | 32 | 10 | $2e^{-4}$ | 1.0 | 0.33 |
| BERT | 16 | 20 | $5e^{-5}$ | 1.0 | 1.0 | 32 | 10 | $3e^{-3}$ | $1e^{-4}$ | 3.5 | 32 | 10 | $2e^{-3}$ | $1e^{-3}$ | 0.33 |
| DistilBERT | 16 | 10 | $5e^{-5}$ | 1.0 | 1.0 | 32 | 10 | $3e^{-3}$ | 0.0 | 4.0 | 32 | 10 | $2e^{-4}$ | $1e^{-6}$ | 0.33 |

Table 6: Hyperparameter Details for the different configurations. The notations represent the following: B - size of bag, E - number of epochs in training, LR - learning rate, $\lambda$ - regularization parameter in the final formulation Eq 6, $\alpha$ - parameter in loss $\mathcal{L}_{TV\star}^{\alpha}$ in Eq 3.

| Model | Dataset | | | | | | | | | |
|---|---|---|---|---|---|---|---|---|---|---|
| | Medical QP | | | | | Financial Phrasebank | | | | |
| | B | E | LR | $\lambda$ | $\alpha$ | B | E | LR | $\lambda$ | $\alpha$ |
| Longformer | 16 | 10 | $3e^{-5}$ | 1.0 | 1.0 | 16 | 10 | $2e^{-4}$ | $1e^{-2}$ | 2.5 |
| RoBERTa | 16 | 10 | $3e^{-3}$ | 1.0 | 5.0 | 16 | 10 | $2e^{-4}$ | $1e^{-2}$ | 2.0 |
| BERT | 16 | 10 | $3e^{-5}$ | $1e^{-2}$ | 0.5 | 16 | 10 | $2e^{-3}$ | 0.1 | 1.0 |
| DistilBERT | 16 | 10 | $3e^{-3}$ | 1.0 | 0.33 | 16 | 10 | $3e^{-3}$ | $1e^{-2}$ | 4.0 |

Table 7: Hyperparameter Details for Medical Questions Pairs and Financial Phrasebank datasets.