# OpenReview forum: "Learning under Label Proportions for Text Classification"
_EMNLP/2023/Conference — EMNLP 2023 Findings_

### Official Review · Reviewer_i5AH · 2023-08-05

**Typos Grammar Style And Presentation Improvements:** 1. Motivation for LLP

The authors mo…
**Soundness:** 3

**Excitement:**

3: Ambivalent: It has merits (e.g., it reports state-of-the-art results, the idea is nice), but there are key weaknesses (e.g., it describes incremental work), and it can significantly benefit from another round of revision. However, I won't object to accepting it if my co-reviewers champion it.

**Paper Topic And Main Contributions:**

The paper proposes a new loss function for training deep learning classifiers in the Learning under Label Proportions (LLP) setup. The proposal is accompanied by a PAC learnability theorem. The paper presents experimental evaluation and comparison against two related approaches, demonstrating significant improvement on 5 text classification dataset.

I would like to note that except for the evaluation on text classification datasets, the paper presents a general deep learning result that is not specific to NLP.

**Questions For The Authors:**

Please give your comments on "Reasons to Reject"

A. Motivation for the approach - how to justify your loss function if the mentioned properties are not needed for a good performance?

B Do you think that the results on bag size of 16/32 give a good estimation of how your method would work in practice?

C Please give the lacking details of experimental setup mentioned above.


**Reasons To Accept:**

- new proposal of a loss function for LLP with a theoretical background
- good experimental results, simple ablation study
- relatively clearly written

**Reasons To Reject:**

1. Motivation for the approach

The authors claim in Section 3 that previously used LLP loss functions lack some properties that can lead to "sub-optimal parameter values post training". The first lacking property is that the loss functions are "not upper bounded". However, the cross-entropy loss function, which is commonly used in deep learning, is also not upper bounded and it does not seem to be a problem in practice. Cross-entropy loss is also not symmetric and not robust (in the sense of the authors). Therefore, the claim that these properties are really needed for more optimal training deserves some citation and discussion.

The authors also claim that their formulation retains "outlier robustness", but this claim is not verified in the experiments.

2. Experiments

My main concern is the practicality of the experimental setups with small bag sizes such as 16/32. The related work tests much larger bag sizes, e.g. (Nandy et al., 2022) has bags with an average of >500 examples, (Liu et al., 2021) uses 128. In addition, the authors evaluate their algorithms on a simpler scenario where the examples change their bag assignment in each evaluation (shuffling the data). I assume that in most practical NLP scenarios the bags would be static.

Some details of the experimental design are missing. For example, are the reported results calculated using cross-validation? Are they averaged over several repetitions? How many? The authors have tuned the lambda and alpha parameters of their method, which seem to be important for the final result (Fig. b,c) - how exactly were they chosen, e.g. what is the range of possible values considered?

The authors evaluate LLP on an ideal case where the proportions from the training data sets are the true proportions and do not consider a more practical variant where there is some noise in the true proportions.



**Reproducibility:**

4: Could mostly reproduce the results, but there may be some variation because of sample variance or minor variations in their interpretation of the protocol or method.

**Reviewer Confidence:**

3: Pretty sure, but there's a chance I missed something. Although I have a good feel for this area in general, I did not carefully check the paper's details, e.g., the math, experimental design, or novelty.

---

> ### Author Rebuttal · Authors · 2023-08-26
>
> We thank the reviewer for the constructive comments and suggestions. We address the questions and concerns in order below:
>
>
> **Q1**: Motivation for the approach -
> **A1**: We first note that these claims are implicitly supported by the results across the tables and figures in the paper. Second and more importantly, the boundedness property helps bridge the gap between the theory and the experiments. A substantial chunk of work in the learnability literature requires bounded losses (and focus on losses such as “0-1 loss”) and similar literature for “cross-entropy” type losses in not deep. This claim is highlighted in line 932 of the appendix in the proof as well. Similarly, the robustness property holds key importance in the light of Lipschitzness. The popular book of [1] provides a thorough discussion of these topics in more detail.
>
> **Q2**: Discussion of the bag size and shuffling of data at each epoch -
> **A2**: We wish to highlight that the primary batch sizes used during tuning of large NLP models typically follow the range between 16 and 32. Since we have followed the protocol to fine-tune the pretrained NLP models, we adhere to these conditions. Furthermore, the experiments in the related LLP literature are focused on CTR and computer vision datasets, wherein it is common to train with a batch size of around 1000. Thus, we think our experimental evaluation is very practical and inline with the large NLP models being deployed in production.
>
> We discuss the practicality of shuffling at each epoch at line 364 under the “Training” part of Section 4.1. More succinctly, since it is highly likely that each instance could be shared across multiple bags, randomly constructing new bags at each epoch does simulate the real-world setup to an extent.
>
> **Q3**: More details of the experimental setup and noise induction -
> **A3**: The details on how we perform cross-validation and tune the hyperparameters are described on line 382 onwards, under the “Training” part of Section 4.1. The range of values considered for $\lambda$ are between $1e^{-7}$ and $1$, with an increment factor of $10$ ($0$ is also considered in a couple of cases as seen from Table 6). For $\alpha$, the corresponding range is between $0.25$ to $5$.
>
> While your suggestion of inducing some noise in the true proportions is indeed a very practical use case, we follow the current setup as an introduction of the problem statement to the NLP community. Furthermore, noise induction also requires much more careful analysis on the amount of noise induced and the corresponding theoretical properties can be harder to analyse in some cases. We will follow up on this suggestion in the future work.
>
> **Q4**: Motivation for LLP -
> **A4**: We appreciate your effort in providing these thoughtfully crafted points. We will first discuss the privacy aspect, followed by the weak supervision:
>
> **Privacy**: First we note that while the data is provided in an aggregated manner, the final aim is to learn an instance level classifier. Thus, it is important to learn a “good enough” classifier (the quantification of “good enough” is a slightly non-trivial matter to resolve here). The recent work of [2] provides very interesting theoretical characterizations of LLP setup. Furthermore, by storing the data in the aggregated manner, one cannot directly infer the truth about a given instance. It is only when we have a well-trained model, we can seek to approximately infer which class a particular instance belongs to. This further makes it safer to transport (transfer) data via a public medium in various scenarios.
>
> **Weak Supervision**: While it may be harder to find such datasets for academic research, we note that such data is available in abundance with tech corporations. In fact, as noted in Nandy et al., 2022, an important use case of LLP arises in the “click through rate” prediction in the advertisement industry, where directly sharing the user level data might be a violation of privacy laws, but sharing on an aggregate level doesn’t directly violate any such constraint. Similar examples where the data belongs to medical and other domains follow.
>
> **Q5**: Other typos and presentation adjustments -
> **A5**: We will correct all the typos and clarify the missing details in the updated manuscript. We appreciate your efforts in pointing out things in such detail.
>
> Regarding the SSC objective in Nandy et al., 2022, we note that their equation 3 is inline and similar to equation 5 in our paper.
> Rest of your suggestions will be accounted for adequately.
>
>
> [1] Shalev-Shwartz and Ben-David : Understanding Machine Learning, From theory to Algorithms (2014)
>
> [2] Learning from Aggregated Data: Curated Bags versus Random Bags - Lin Chen, Thomas Fu, Amin Karbasi, Vahab Mirrokni; (2023)
>
> If the response satisfies your concerns, we hope that you will improve the score to reflect the same. We are happy to answer any more questions.

---

### Official Review · Reviewer_g6p6 · 2023-08-05

**Soundness:** 3

**Excitement:**

2: Mediocre: This paper makes marginal contributions (vs non-contemporaneous work), so I would rather not see it in the conference.

**Paper Topic And Main Contributions:**

The paper investigates the application of the Learning form Label Proportion (LLP) to natural language. In particular, the authors present a new training objective which is a weighted combination of a modified version of total variation distance and self-supervised contrastive loss. The paper presents some theoretical justification and an extensive parameter analysis. The effectiveness of the method is demonstrated by experiments on five well-known datasets and four models.


**Reasons To Accept:**

- The application of LLM to natural language tasks is new and of some interest.
- The proposed method is theoretically justified; properties of the method are discussed.
- Interesting ablation study and result analysis.


**Reasons To Reject:**

- The paper is hard to read and somewhat difficult to follow.
- The motivation is unclear. The authors argue that the LLP setup is relevant for (1) privacy and (2) weak supervision. (1) Privacy: the authors claim that the LLP paradigm is relevant for training on sensitive data as the labels for such datasets are not publicly available. However, the setting proposed in this paper does require gold (and publicly available) labels to formulate the ground truth proportion. If this proportion can be formulated without gold labels, it should be discussed. (2) Weak Supervision: in lines 136-137, the authors mention that the associated label proportions "...provides the weak supervision for training the model". However, weak supervision is a paradigm in which data is automatically labeled with noisy labels using some heuristics and labeling functions. It remains unclear to me in what way this setting is related to the proportion parameter authors use in their work.
- The authors claim it to be one of the preliminary works discussing the application of LLP to NLP tasks. However, I don't see anything NLP-specific in their approach.
- Not all theoretical groundings seem to be relevant to the main topic (e.g., some of the L_dppl irregularities). Additional clarification of their relevance is needed.
- Section 3.3 says the results are provided for binary classifiers only, and the multi-class setting remains for future work. However, one of the datasets used for experiments is multi-label.
- The experimental setting is unclear: does Table 1 contain the test results of the best model? If so, how was the best model selected (given that there is no validation set)? Also, if the proposed method is of special relevance to the sensitive data, why not select a sensitive dataset to demonstrate the method's performance on it? Or LLP data?
- The authors claim the results to be significant. However, no results of significance testing are provided.

**Reproducibility:**

3: Could reproduce the results with some difficulty. The settings of parameters are underspecified or subjectively determined; the training/evaluation data are not widely available.

**Reviewer Confidence:**

4: Quite sure. I tried to check the important points carefully. It's unlikely, though conceivable, that I missed something that should affect my ratings.

**Typos Grammar Style And Presentation Improvements:**

- The font of Figure 1 and Table 1 is too small and hard to read.
- Tables 5 - 7 are listed in Appendix, but there is no reference to the Appendix when referring to them.
- Superfluous brackets in some citations (lines 35-36, 71, 79, 161, 556, 558, 559, ...).
- lines 213, 493: the word "Appendix" is missing a capital letter.
- lines 481, 483, 497: missing full stop after "Eq".

---

> ### Author Rebuttal · Authors · 2023-08-26
>
> We thank the reviewer for the constructive comments and suggestions. We address the questions and concerns in order below:
>
> **Q1**: Unclear motivation -
> **A1**: We understand the concern with the writeup and will update in the subsequent version of the manuscript. We would like to clarify both the points in order:
>     **Privacy**: Since there are no existing datasets that follow the LLP setup in the NLP domain, we have to manually simulate this by aggregating ground truth labels during training. We also highlight that the background literature works - DLLP and LLPSelfCLR both operate similarly.
> In real world scenarios, we typically have access to the aggregated data beforehand. A real world example of this is - when firms like twitter and google gather data from e-commerce platforms like amazon and shopify (oracle in this case), the latter already provide aggregated statistics to the former. Thus the former has to begin training their models using the aggregated data from the get go.
>
> **Weak Supervision**: Here, we have used the broad definition of weak supervision as discussed by Nandy et al. 2022. This helps us remain coherent with the literature. Under this broad definition, we refer to a task which does not provide ground truths at the instance level but rather noisy (as you have pointed out) or aggregated (in this paper) labels, as weak supervision.
> We have discussed this in the introduction, line 56, of the paper.
>
> **Q2**: Nothing NLP-specific in the approach -
> **A2**: As you have already described, we provide one of the preliminary works discussing the application of LLP to NLP tasks. The aim here is to bring the attention of the community to this highly important problem. Furthermore, due to this reason, the baselines used in our comparison are more general purpose. Going forward, we hope to incorporate more NLP specific paradigms into LLP.
>
> **Q3**: Not all theoretical groundings seem to be relevant -
> **A3**: We note that the section describing irregularities of $L_{DLLP}$ is what primarily motivated our novel loss function formulation. Thus, we believe that adding some theoretical characterization of these properties along with the main result in theorem 2 leads to a more complete work. Furthermore, properties such as #2 (robustness) and #5 (lipschitz smoothness) have high significance in machine learning research, which our formulation exhibits.
>
> **Q4**: Discussion on Section 3.3 -
> **A4**: We wish to clarify that Section 3.3 solely focuses on the theoretical result. For ease of discussion and succinctness in the theory part, we provide theorem 2 in the light of binary classifiers, but is well extendable to multi-class settings.
> The main experiments do contain both binary and multi-class (with more than 2 labels) settings for the completeness of practical research purposes.
>
> **Q5**: Unclear experimental setting -
> **A5**: We note that the main results in Tables 1 and 2 focus on the comparison between our loss formulation and the baselines. This is to exhibit that the proposed loss function is better suited for the LLP setting across a broad range of configurations. To demonstrate this, we selected a diverse suite of models and datasets. Thus each model-dataset combination can be observed on a standalone basis and it is noted that our formulation performs better in most of the cases. The discussion regarding dataset selection follows from the response of **Q1** and **Q2** above.
>
> **Q6**: No significance testing is done -
> **A6**: Since we found that the results across all the configurations are fairly stable and reproducible, we provide the mean value of the numbers. This is inline with the literature (corresponding to fine-tuning of large scale pretrained models) where the variance in results are typically of no significance, as observed  across various types of datasets.
>
> We will adjust the presentation style and correct the typos as per your suggestion in the updated manuscript. We appreciate your efforts in pointing out things in such detail.
>
> If the response satisfies your concerns, we hope that you will improve the score to reflect the same. We are happy to answer any more questions.

---

### Official Review · Reviewer_qeAx · 2023-08-07

**Soundness:** 4

**Excitement:**

4: Strong: This paper deepens the understanding of some phenomenon or lowers the barriers to an existing research direction.

**Paper Topic And Main Contributions:**

This paper introduces a robust objective function for the learning from label proportions (LLP) setting. Unlike the traditional KL divergence utilized in the DLLP work by Ardehaly and Culotta (2017), the new objective function applies the total variation distance, which is a tight lower bound of the KL. The paper has provided theoretical results, based on structural risk minimization framework, and empirical results,  on a number of text classification datasets, to support the proposed new formulation.

**Reasons To Accept:**

I've found the LLP setting quite interesting, especially for data privacy applications. The paper has also adequately supported the newly proposed loss function based on the total variation distance, from both theoretical and empirical perspectives. Overall, it is a good quality paper.

**Reasons To Reject:**

I don't have any major reasons for rejection. However, I have a few comments:

- While the evaluation metrics (weighted precision, recall and F1) follow prior works, to be self-contained, it would be beneficial to precisely define these metrics, especially considering (a) how important evaluation is and (b) different versions of these weighted metrics exist.

- The base models used in the experiments predominantly belong to the BERT family. Apart from the encoder models, have you considered using decoder models as the base models?

**Reproducibility:**

3: Could reproduce the results with some difficulty. The settings of parameters are underspecified or subjectively determined; the training/evaluation data are not widely available.

**Reviewer Confidence:**

3: Pretty sure, but there's a chance I missed something. Although I have a good feel for this area in general, I did not carefully check the paper's details, e.g., the math, experimental design, or novelty.

---

> ### Author Rebuttal · Authors · 2023-08-26
>
> We thank the reviewer for the constructive comments and suggestions. We address the questions and concerns in order below:
>
> **Q1**: Regarding the evaluation metrics -
> **A1**: We agree with your comment and will provide a detailed description of the metrics in the updated manuscript.
>
> **Q2**: Use of decoder models -
> **A2**: Since the problem setting is mostly concerned with classification, we used the encoder family as most of the classification tasks are performed using the final layer embeddings of the encoder component. These are then typically passed through a linear layer and subsequently into the loss function.
> However, we have taken your comment into note and are looking into the literature for any relevant decoder model that can be leveraged (we will update the manuscript accordingly).

---

### Meta-Review · Area_Chair_3yML · 2023-09-19

**Recommendation:** 4

**Metareview:**

In words of one of the reviews:
"The paper investigates the application of the Learning form Label Proportion (LLP) to natural language. In particular, the authors present a new training objective which is a weighted combination of a modified version of total variation distance and self-supervised contrastive loss. The paper presents some theoretical justification and an extensive parameter analysis. The effectiveness of the method is demonstrated by experiments on five well-known datasets and four models."

A clear strength of this paper is the out-of-the-trodden path setting which opens up interesting new research questions, as well as - arguably (see below) practical applications. The contributions of this paper are both theoretical as well as empirical.

The main criticisms are about the practicality of the setting - having proportion of levels but not fine-grained ones. While this seems like a reasonable situation, it would be arguably more compelling if a real use-case could be mentioned (if not used). There are although concerns about the clarity of the writing - the authors are encouraged to take those comments to heart.

---

### Decision · Program_Chairs · 2023-10-07

**Decision:**

Accept-Findings

**Comment:**

In words of one of the reviews:
"The paper investigates the application of the Learning form Label Proportion (LLP) to natural language. In particular, the authors present a new training objective which is a weighted combination of a modified version of total variation distance and self-supervised contrastive loss. The paper presents some theoretical justification and an extensive parameter analysis. The effectiveness of the method is demonstrated by experiments on five well-known datasets and four models."

A clear strength of this paper is the out-of-the-trodden path setting which opens up interesting new research questions, as well as - arguably (see below) practical applications. The contributions of this paper are both theoretical as well as empirical.

The main criticisms are about the practicality of the setting - having proportion of levels but not fine-grained ones. While this seems like a reasonable situation, it would be arguably more compelling if a real use-case could be mentioned (if not used). There are although concerns about the clarity of the writing - the authors are encouraged to take those comments to heart.